# The Content of Selected Bioactive Compounds in Wines Produced from Dehydrated Grapes of the Hybrid Variety ‘Hibernal’ as a Factor Determining the Method of Producing Straw Wines

**DOI:** 10.3390/foods11071027

**Published:** 2022-04-01

**Authors:** Barbara Kowalczyk, Monika Bieniasz, Anna Kostecka-Gugała

**Affiliations:** 1Department of Ornamental Plants and Garden Art, Faculty of Biotechnology and Horticulture, University of Agriculture in Krakow, Al. 29-Listopada 54, 31-425 Kraków, Poland; 2Department of Horticulture, Faculty of Biotechnology and Horticulture, University of Agriculture in Krakow, Al. 29-Listopada 54, 31-425 Kraków, Poland; bieniasz.monika@urk.edu.pl; 3Department of Plant Biology and Biotechnology, Faculty of Biotechnology and Horticulture, University of Agriculture in Krakow, Al. 29-Listopada 54, 31-425 Kraków, Poland; anna.kostecka-gugala@urk.edu.pl

**Keywords:** hybrid cultivars, *passito* wine, straw wine, polyphenols, antioxidant capacity, antiradical capacity

## Abstract

Sweet wines are appreciated worldwide; many are produced by fermenting the must of dehydrated (semi-dried) grapes, using methods that vary from region to region. The aim of this study was to evaluate the basic chemical and oenological characteristics of wines obtained by three technologies of production. The wines were made from a hybrid cultivar ‘Hibernal’, grown under cool climate conditions. ‘Hibernal’ is a hybrid variety. This ‘Hibernal’ variety is widely cultivated in central and eastern Europe, where it is of great economic importance. Wines produced from this variety are popular in local markets. In comparison with the production of varieties belonging to *Vitis vinifera*, a very small percentage of the ‘Hibernal’ variety is cultivated. The methods used in the experiment for the production of wines were: classical method in the Italian *passito* style, modification of the *passito* style with a seven-day maceration of grapes, and a method of production in the Tokaj wine style at five Puttonyos. Basic chemical parameters, acid profile, total phenolic content, antioxidant and antiradical capacities, and quantitative analysis of selected polyphenols was performed. The sensory features and quality of the wines was assessed using a sommelier analysis based on The Wine & Spirit Education Trust guidelines. The results indicated that the seven-day maceration of the dehydrated grapes resulted in the highest polyphenol content, as well as the largest antioxidant and antiradical contents. The oenological evaluation of wines produced by the Tokaj method and Italian *passito* method with seven-day maceration found that the wines were appreciated due to their rich taste, flavor, and overall quality. The present study confirms the promising opportunities to obtain special sweet wine with a valuable composition and oenological characteristics in regions with cooler climates.

## 1. Introduction

Sweet dessert wines are very popular in the world. They are produced using various technologies primarily based on the grape dehydration process. This treatment is aimed at increasing sugar concentration in the must, and can be carried out using several methods, such as cryoconcentration, for German and Canadian Eiswein, *Botrytis cinerea* infection for Tokaj wines, stopping fermentation by distillate addition for Porto or Madeira wines, or by partial grape drying. The latter method is the most common and has been used for centuries. Several countries have a long tradition in producing these types of wines. For instance, Italy’s production of Vino Passito and Vin Santo, Austria’s production of Strohwein, France’s production of Vin de Paille, and Spain’s production of Pedro Ximenez. The old methods were based on spreading the grapes on mats and drying them in the sunshine. At present, modern dryers with a forced warm, dry air convection are mostly used, thus avoiding problems associated with fungal growth and ochratoxin, contamination, or insect problems [1,2]. The winemaking process itself also provides many problems. A high sugar concentration and increased alcohol content create osmotic stress for yeasts, and only some of them are able to carry out the fermentation process. The end products are wines with a very distinctive taste and aroma, which are often more complex than standard red or white wines made from ordinarily ripe grapes [3,4]. Mostly, they are sweet to very sweet, capable of a long life, usually deep golden in colour, with a viscous appearance. In general, the nose and palate is met with a complex, alluring blend of peaches, dried apricots, and marmalade, with flavours of almonds and honey. The intense mouthfeel is balanced by a clean, fresh, and very long finish of dried apricots [3,4,5].

In Poland, in recent years, there has been an increase in the number of vineyards where mostly hybrid varieties are grown. *Vitis vinifera* strains are cultivated only in the west of the country. The whole country is located in the cool climate zone, and it is rarely possible to produce wines with higher residual sugar contents under natural conditions. Generally, it is only possible to harvest grapes with a relatively low extract (usually 17–23° Brix), although these can give good, fragrant dry, or semi-dry wines with a balancing acidity. [5]. The only way to obtain dessert wines is to use special production methods. The climate that prevails in Poland should favor the production of ice wines, however, there is no guarantee that a frosty winter will happen every year. At the turn of 2020 and 2021, only 500 L of ice wine were produced, which accounted for 0.03% of the total wine production in Poland. Straw wines (i.e., wines made of semi-dried grapes being dehydrated indoors) are a safer solution for producers, as they are independent of weather conditions [1].

A number of studies provide evidence for the beneficial effects of grapes on human health, both in their basic and modified forms. A special role is attributed to polyphenolic compounds contained in grapes, which greatly contribute to the prevention of cardiovascular diseases and cancer [6,7,8]. Most of them are found in red wines, due to their production technology. Maceration, together with skins and seeds, results in the extraction of their ingredients into the must. For white wines, the process is slightly different, as the must is fermented directly after pressing, reducing valuable compound contents such as polyphenols by up to 10-fold. For winemakers, polyphenols are important, primarily for the colour and taste of wine [9,10]. Phenolic compounds contained in unfermented grapefruits are not fully assimilated by the human body because they occur as large polymeric complexes. Ethanol formed during the fermentation process is a good solvent for phenolic compounds, hence facilitating their extraction into wine [11].

One grape variety (i.e., hybrid variety ‘Hibernal’) was used in the experiment, but three different production methods were applied. In commercially available wines, one can also find similar examples of different technologies being used during production in order to produce different wine characteristics (e.g., Amarone della Valpolicella and Recioto della Valpolicella, where the maximum sugar content of the former is 12 g/L, and over 100 g/L of the latter) [12]. In this study, three wines with different styles and compositional parameters were obtained. Their variability was mainly due to different production technologies (i.e., classic for straw wines: grape drying (wine *A*), grape drying and maceration (wine *B*) and wine refermentation using dehydrated grapes (wine *C*)).

During the long preliminary procedure (i.e., drying, fermentation, and maturation to the final product), straw wine is enriched with several compounds, similarly to classic wines. In wines made from dried grapes, substances dissolved in water, mainly sugars, but also acids, are concentrated [13,14,15].

By applying the red wine production method to straw wine making, a product with a higher concentration of polyphenols, that is comparable to red wines, can be obtained. The objective of this study was to evaluate the production methods of straw wines, considering their oenological characteristics, as well as the content and proportion of essential health-promoting compounds. The results will help improve the quality of straw wines produced under cool climate conditions.

## 2. Materials and Methods

### 2.1. Biological Material

The research was carried out for three types of straw wines produced in 2015, from the hybrid ‘Hibernal’ variety (Figure 1A), derived from the ‘Seibel 7053’ (Chancellor), and ‘Riesling’ cultivars.

Straw wines used for the comparison were made of grapes from the vineyard ‘Garlicki Lamus’, located nearby to Krakow, in the south of Poland (50°08′29.4″ N 19°55′50.7″ E).

The collected fruits were fully ripe, without any damage, infection, mold, or rot. The process of dehydrating the grapes, which lasted six weeks, took place in a dry and well-ventilated room, in openwork boxes (Figure 1B). Two fans were used to force air circulation. The temperature during drying was kept at about 18 °C. During the entire process, the grape clusters were inspected, and any damaged or infected fruit were immediately removed (Figure 1C). The basic chemical parameters of fresh and dried grapes are summarized in Table 1.

The straw wines used for the comparison were made using three methods:

A: The 50 kg semi-dried grapes were pressed, then the must (about 20 L) was fermented (the classical straw wine method);

B: The 50 kg semi-dried grapes were ground and macerated for 7 days at 15 °C. Then, the free-run juice was poured off and the remaining grapes were pressed. The must was combined with the free-run juice and subjected to a further fermentation process—about 20 L (the classical method with fermentative maceration);

C: The 3 kg semi-dried grapes were poured with 20 L of fresh ‘Hibernal’ wine in a ratio of 1 kg per 7 L and subjected to refermentation. The fresh wine had a residual sugar content of 6.7 g/L and an acidity of 6.05 g/L (the Tokaj method). Fermentation of the young wine was carried out with ZYMAFLORE^®^ VL1 yeast made by Laffort. IOC Bayanus yeast was used for refermentation at 50 g/hl.

Active dried yeast IOC Bayanus produced by Institut Oenologique de Champagne (*Saccharomyces cerevisiae bayanus*) was used to produce and referment the wines used in the research. After the supernatant was decanted, the wine was allowed to clarify. The wines were aged at 10 degrees Celsius and 75% humidity.

### 2.2. Physicochemical Analyzes

#### 2.2.1. pH, the Total Acidity, the Extract Content

The pH and the total acidity of the wine were determined with a Janway 3020 pH meter. Titration with NaOH was performed to pH = 7.0. The analysis was performed in triplicate. The results were expressed in g × 100 g^–1^ of tartaric acid equivalents. For determination of the extract content, an electronic ATAGO Digital PR-101a refractometer (Tokyo, Japan) with a range of 0.0–45.0 °Brix was used. The measurement was performed in triplicate and the results were expressed in °Brix.

#### 2.2.2. Ethanol Content

The determination of the ethyl alcohol content of the straw wine samples were tested according to the OIV reference method (OIV-MA-AS312-01A). Briefly, 100 mL of wine with the addition of 10 mL of 2 M Ca(OH)_2_ was distilled at 20 °C. Calcium hydroxide is necessary to avoid the presence of volatile acids which could affect the ethanol quantification. Distillation was stopped when approximately 75% of the original volume was reached, and after 20 min, pure water was added to the total volume of 100 mL. The ethanol content was measured with an oscillating type densimeter (Handheld DensityMeter, Densito, Mettler-Toledo, Columbus, OH, USA).

#### 2.2.3. Acid Profile

The content of organic acid anions was investigated with an EA 100 capillary isotachoresis (ITP) analyzer (Villa Labecos. r.o. Spisska Nova Ves, Slovakia). The apparatus was equipped with two capillaries: a pre-separation capillary (90 mm × 0.8 mm ID) and an analytical capillary (90 mm × 0.3 mm ID). The initial separation was carried out at an AC current of 250 µA, while the separation in the analytical capillary column at 60 µA during the initialization step and at 50 µA during the detection step. The lead electrolyte (LE) was 10 mM HCl with 0.1% HEC (Hydroxyethyl Methyl Cellulose) and 10 mM β-alanine. The final electrolyte was a 0.01 M caproic acid solution [16,17]. All the reagents used for ITP analysis (i.e., β-alanine, caproic acid, tartaric acid, citric acid, malonic acid, succinic acid, lactic acid, and acetic acid, as well as hydrochloric acid) were from Sigma–Aldrich (Burlington, MA, USA).

#### 2.2.4. Total Phenolic Content (TPC)

The number of phenolic compounds in the extracts was determined based on the reaction with the Folin–Ciocalteu reagent. The wine sample (0.25 mL) was mixed with 0.25 mL of 25% Na_2_CO_3_, 0.125 mL of the Folin–Ciocalteu reagent (Sigma–Aldrich, diluted twice with water prior to the analysis), 2.25 mL of water, and then incubated for 15 min. The absorbance was measured at 760 nm (JASCO V-530 UV–Vis spectrophotometer). The final results were expressed as mg of gallic acid (Sigma–Aldrich) per L (gallic acid equivalents, GAE mg L^−1^) [17,18].

#### 2.2.5. Antioxidant Capacity—A FRAP Assay

The FRAP (ferric reducing antioxidant power) assay is based on the reduction of the ferric–tripyridyl-*s*-triazine (Fe^+3^–TPTZ) complex to its ferrous (Fe^2+^) derivative. The FRAP working solution was freshly prepared by mixing together 300 mM acetate buffer (pH 3.6), 10 mM TPTZ (Sigma–Aldrich) in 96% ethanol, and 20 mM FeCl_3_ (10:1:1, *v:v:v*). Then, 3 mL of the FRAP working solution were mixed with 0.1 mL of the wine sample and 0.3 mL of water. The absorbance was measured at 595 nm after 5 min. The results were expressed as µmol Trolox (6-hydroxy-2,5,7,8-tetramethylchroman-2-carboxylic acid; Sigma–Aldrich) per L (Trolox equivalents, TE µmol L^−1^) [19].

#### 2.2.6. Antioxidant Capacity—A CUPRAC Assay

The CUPRAC (cupric ion reducing antioxidant capacity) assay is based on the measurement of the utilization of copper (II)-neocuproine as a chromogenic oxidizing agent. Briefly, 1 mL of 10 mM CuCl_2_, 1 mL of 7.5 mM neocuproine (Sigma–Aldrich) in 96% ethanol and 1 mL of 1 M NH_4_Ac buffer, pH 7.0, were mixed with 0.25 mL of the wine and 0.8 mL of distilled water. The absorbance was measured at 450 nm after 15 and 30 min. The results were expressed as µmol Trolox (Sigma–Aldrich) per L (TE µmol L^−1^) [20,21].

#### 2.2.7. Radical Scavenging Capacity (RSC)—A DPPH Assay

The radical scavenging capacity of extracts was tested following the reduction of a synthetic, stable free radical 2, 2-diphenyl-1-picrylhydrazyl (DPPH). The colorimetric method enables the measurement of absorbance changes to DPPH solution at 517 nm as a result of the antioxidant activity of the sample. Briefly, 2.8 mL of 0.1 mM DPPH (Sigma–Aldrich) solution in 96% ethanol was mixed with 0.2 mL of the wine sample. The DPPH absorbance was measured after 10 min. The RSC results were expressed as µmol Trolox (Sigma–Aldrich) per L (TE µmol L^−1^) [20].

#### 2.2.8. Polyphenol Analysis

High performance liquid chromatography (HPLC) (Shimadzu LC-10AS chromatograph equipped with a C18 RP column and SPD-10AV UV-Vis detector) was used to identify phenolic compounds in wine. Signal detection was set at 265 and 325 nm. Chromatographic separation was performed at 33 ± 1 °C using the following solvents: (A) water (Sigma–Aldrich) with acetic acid (0.1%), (B) methanol (Sigma–Aldrich, ultrapure) with acetic acid (0.1%), and use of a solvent gradient: 90% A, 10% B for 20 min; 75% A, 25% B for 30 min; 65% A, 35% B for 40 min; 55% A, 54% B for 50 min; 50% A, 50% B for 60 min; 30% A, 70% B for 62 min; 100% B for 80 min; 80% A, 10% B up to 85 min. The flow rate was 1 mL/min. Identification of phenols was based on the retention times of vanillic, syringic, *trans*-Cinnamic acids, quercetin (Sigma–Aldrich), and (+)-catechin (LGC Standards), at 265 nm, and chlorogenic, caffeic, *p*-coumaric, ferulic acids, *trans*-resveratrol (Sigma–Aldrich), at 325 nm. Before separation, the samples A and C were diluted twice, and the sample *B* was diluted four times, all with methanol (Sigma–Aldrich, HPLC gradient, ultra pure).

### 2.3. Organoleptic Analyses

All wines were subjected to organoleptic analysis by a group of 30 respondents in the age group of 22–55. Study participants were free of upper respiratory tract infections. All respondents were amateur wine consumers, and they received basic training in organoleptic evaluation of wines based on The Wine & Spirit Education Trust (WSET) materials. The training was conducted by two persons with WSET Level II sommelier certifications. During the training, 64 aromas from LE Nez du Vin (Jean Lenoir, Provence, France), as well as solutions for taste blindness (flavors: sour, sweet, bitter, salty, and umami) were utilized. Wines were evaluated on “wine evaluation sheets” which were compiled based on the WSET materials. The scores presented on the sheets were intended to illustrate the quality rating of individual wines. The Table 2 includes answers that were marked by at least 60% of the panellists in the survey.

### 2.4. Statistical Analysis 

The collected results were processed using a one-way analysis of variance. Tukey’s test was used to assess the significance of differences between the means, with a significance level of α = 0.05. All the statistical calculations were performed using Statistica 13 computer software.

## 3. Results and Discussion

The physical and chemical parameters of the obtained straw wines were determined in the experiment. The first analyzed parameter was the pH of the juice, which is a factor that influences must and wine stability [22]. The initial juice pH in the grapes after drying increased from 3.14 to 3.31 (Table 1). This process is observed when the juice is concentrated, but also during extended maturation [23]. During fermentation, the pH value increased to 3.91 for sample *A*, and the statistically lowest and most desired pH was achieved for sample *C* (Table 3). Aponte and Blaiotta [24] analyzed the passito wine ‘Moscato di Saracena’, and recorded a pH range from 3.1 to 3.93, which was similar to the values obtained in the present study. Bondada et al. [23] confirmed that pH was not strictly correlated with acidity (TA), and pH increase did not always show the same trend, which was also reflected in the results of this study.

The main organic acids in wines, such as tartaric acid, citric acid, and malic acid come from fruits; others are fermentation products (i.e., lactic acid, succinic acid or acetic acid), and affect wine sensory characteristics such as colour, and its microbiological stability [25,26]. The titratable acidity of wines in the experiment was in the range of 8.18–10.34 g/L; it was statistically the highest in sample B, which was probably due to the high initial acid content in the macerate which was then fermented (Table 3). Climatic conditions are key factors in determining grape maturity and maturity-related parameters, such as sugar content and total grape acidity. In classic wines, it is assumed that total acidity should be between 5 and 12 g/L [27]. Croce et al. [12] studied 302 samples of Italian wines made from dried grapes. The acidity recorded by these authors ranged between 3.73 and 11.31 g/L. The values obtained in this experiment were at the upper end of this range.

Wines made from dried grapes can vary in typology, with sugar concentrations reaching even 600 g/L. Wines in the present study could be divided into two groups in terms of sugar content: wines A and B belonged to very sweet wines with a sugar content higher than 150 g/L, and wine C belonged to the group of sweet wines, with sugar in the range of 60 to 150 g/L. Domizio & Lencioni [28] demonstrated a high diversity among *passito* styles. They summarized that Tuscan wines (Vin Santo) had residual sugar content between 10 and 250 g/L. Their observations indicated that recent trends in consumption have prevailed toward slightly sweet and sweet wines. Laureati et al. [29] also described the production of traditional Vin Santo, made from three varieties: Trebiano, Malvasia and Grechetto. The wines were divided into three types: semi-sweet wines (10–50 g/L residual sugar), slightly sweet wines, and sweet wines (up to 100 g/L residual sugar content). The described production technology for Vin Santo was very similar to that used by the authors for sample A; however, a residual sugar in the experiment was determined at 182 g/L, which classified it as a very sweet wine (Table 3). By reducing the degree of dehydration, and thus sugar level in the must, it is possible to match the style of Polish wines to traditional Italian wines. Giordano et al. [30] described obtaining Passito di Pantelleria DOC wines using a base wine and refermentation to sweeten the product, yielding wines containing 200 to 340 g/L residual sugar. In the present study, the authors used the same method for wine *C* but obtained a lower sugar concentration (i.e., 69 g/L). Such a result was due to the low residual sugar content in the base wine and in the dehydrated grapes. No reports were found in the literature on using dried grapes and the maceration method to produce wines, which was applied in sample B. Such a method is sometimes used to produce traditional white wines (very short maceration), and almost always to produce red wines (longer maceration).

The yeast used in the study (i.e., *S. bayanus*), is applied to resume fermentation, refermentation, fermentation of meads, and production of sparkling beverages [31,32]. In this case, the fermentation in two samples (wine *A* and *B*) stopped at the maximum alcohol concentration of 18% vol, whereas the alcohol concentration in sample *C* was 15% vol (Table 3). Croce et al. [12] recorded alcohol content in *passito* wines ranging from 10.18 to even 20.52% vol. Similar results were obtained by Domizio and Lencioni [28], who examined 63 Tuscan Vin Santo wines and determined that the alcohol concentration was in the range of 10–19% vol. They divided Vin Santo into three styles depending on residual sugar and alcohol content: dry style (16–19% vol alcohol and 10–50 g/L sugar), slightly sweet and sweet style (14–16% vol alcohol and up to 100 g/L sugar), and very sweet (14–16% vol alcohol and up to 250 g/L sugar). Relating the results of the present study to the above division, wine *C* can be classified as a slightly sweet style, whereas wines *A* and *B* would be classified as very sweet in terms of sugar content, and as having a dry style in relation to alcohol concentration.

The current study analyzed the content of major organic acids by isotachophoresis (ITP) (Table 3). This method is used for electromigration separation and allows the analysis of mixtures of ionic substances [33]. In the experiment, sample *B* had the highest content of tartaric, malic and succinic acids, and together with wine *A*, also citric acid. The lowest contents of tartaric, citric, malic, succinic, and acetic acids were measured in sample *C*. Aponte and Blaiotta [24] obtained similar values as the authors for wine *A*: tartaric acid content—2.56–2.65 g/L, citric acid—0.54–0.60 g/L, malic acid—2.16–2.11 g/L, succinic acid—1.72–1.89 g/L, acetic acid—0.79–1.51 g/L, and in only one case, lactic acid—0.84 g/L. Giordano et al. [30] studied three Italian *passito* wines and acquired similar contents of the aforementioned acids to those obtained in the present study. Wine *C* proved to be the most similar to Passito di Panteleria DOC in terms of organic acid content; the production method of both wines was also very similar. In the experiment, as in a study by Aponte and Blaiotta [24], wine *A* statistically had the highest acetic acid content at 1.5 g/L, and fermentation dynamics and acetic acid production were also evaluated. The maximum concentration of acetic acid formed in wines from dehydrated grapes has been defined in several countries; for example, the maximum value of acetic acid in Canadian Eiswein is 2.1 g/L [4,34]. In Europe, the standard for white and rosé wines is 1.08 g/L, and 1.8 g/L for red wines. An elevated acetic acid content in table wines usually indicates a spoilage process; however, in the case of wines made from dried grapes, it is responsible for retaining a redox balance when responding to osmotic stress induced by high sugar levels [34].

The total polyphenol content of wine is influenced by winemaking techniques, namely maceration, fermentation, and aging [35]. In white wines, it is assumed to range from 100 to 400 mg/L, in rose wines from 400 to 800 mg/L, and in red wines from 1000 to 2000 mg/L [36]. The straw wines used in the present study contained 907.5 to 3748.5 mg of gallic acid equivalents per liter (mg GAE/L); sample *B* (i.e., the one in which the dried grapes were macerated) was the richest in phenolic compounds (Table 4). Many authors have shown that the maceration process leads to an increase in total polyphenols due to their high concentration in the skins and seeds. Their greatest increments occur at the end of maceration, when the produced alcohol destroys the skin lipid layer that protects the seeds [37,38]. For wines *A* and *C*, the concentration of polyphenols was over 70% lower than for sample *B* in the current study. Fhurman et al. [39] studied the effect of skin contact with grapes on the polyphenol content and antioxidant capacity of white wine. During maceration, which lasted from 2 to 18 h, the latter authors noted a gradual increase in the concentration of polyphenols (up to 41%). They also analyzed the effect of fresh grape maceration in alcohols of different concentrations (2–18% vol). The 18% vol alcohol resulted in the highest polyphenol extraction, 60% higher than in the wine without any added alcohol. In the present study, wine *A* had no contact with skins and seeds, whereas polyphenols present in the dried grapes in wine *C* were dissolved in the base wine. A lower alcohol concentration, compared to other samples, may not have been sufficient to fully extract polyphenols from dried grapes. Panceri and Bordignon-Luiz [26] analyzed the effect of grape dehydration, among others, on the chemical composition of young wines from dark Merlot and Cabernet Sauvignon varieties using a five-day maceration after 22 months of maturation. They determined total phenolic content (TPC, in GAE) in the range of 1221.78–1588.50 mg GAE L^−1^ (i.e., they obtained values similar to samples *A* and *C* after 5 years of maturation). Much lower TPC values (an average of 249 mg/L) were obtained by Loizzo [35], who analyzed *passito* from Saracena that was aged between one to four years; however, he expressed the results as (+)-catechin equivalents. The production of this wine was slightly different from the one used in this study. Fresh must was obtained from overripe grapes and dehydrated grapes were added to it. During the wine aging process, the phenolic composition changes quantitatively and depends on the type of wine and its storage conditions [40]. The conducted analysis confirmed the differentiation of the total polyphenol content depending on the production method.

The antioxidant properties of the wines were evaluated by FRAP and CuPRAC assays. FRAP is commonly used in many studies of antioxidative compounds, CuPRAC is less known, but is considered to be more sensitive [41]. The analysis of the obtained values showed that sample *B* revealed the strongest antioxidant properties. The results from FRAP and CuPRAC analyses are well correlated with TPC [42], and as shown above, sample *B* was richest in phenolic compounds. This also confirmed that the phenolic compounds in straw wine *B* were characterised by a higher antioxidant potential.

Studies of of radical scavenging capacity (RSC), defined as the ability to quench the DPPH free radical, showed that sample *B* was 2.7 times more effective as an antiradical agent than sample *A*, and 2.5 times more effective than sample *C*. In red wines, RSC is usually more than 10 times higher than in white wines [43]. Panceri et al. [44] showed the RSC in the range of 2.67–4.67 mM/L (TE). Li et al. (2009) [45] tested 24 Chinese classic wines and reported activity for red wines ranging from 4.19 to 21.30 mM/L (TE), for rosé wines from 1.40 to 3.41 mM/L (TE), and for white wines 0.08 to 1.12 mM/L (TE). Noting the results of the antiradical capacity with the literature data, it could be observed that the values obtained for samples *A* and *C* corresponded to the characteristics of typical white wine, compared with the red wines of sample *B*.

The polyphenol composition of wine samples was analyzed using high-performance liquid chromatography (HPLC) with the use of standards (Table 5). Polyphenols were identified as flavonols ((+)catechin, quercetin), hydroxybenzoic acids (vanillic acid and syringic acid), hydroxycinnamic acids (*p*-coumaric acid, caffeic acid, chlorogenic acid, ferulic acid, *trans*-cinnamic acid), and stilbene (*trans*-resveratrol). Two of the phenolic compounds (e.g., syringic acid; (+)-catechin), belonged to the chemical markers of ‘Hibernal’ white wine [46].

(+)-Catechin content of the wines in the present experiment was in the range of 37.82–64.29 mg/L, with only wine *C* having statistically lower contents than wines *A* and *B* (Table 5). Loizzo [35] recorded 71.7 mg/L of catechins in fresh Passito of Saracena wine. Avizcuri-Inac et al. [47] analyzed the chemical and sensory characteristics of mature sweet wines obtained using different techniques, and determined that catechins in white wines range from nd.–0.12 mg/L, and in red wines from nd.–0.17 mg/L. In twelve white commercial wines, Han et al. recorded catechins in the range of 3,57–16,87 [48]. Rozdrigez-Delgrado et al. [49] analyzed 55 fresh red wines from the Canary Islands and obtained values ranging from 17.70 to 30.77 mg/L. Panceri et al. [44] also analyzed catechin contents in red wines and acquired values ranging from 5.82 to 18.19 g/L. Low catechin content is usually found in older wines and it is related to their polymerization Tannins that contribute to dark colour and astringency of wine mainly belong to condensed tannins (or procyanidins) derived from seeds and skins [35,50]. The main forms are polymers of flavan-3-ols ((+)-catechin, (−)-epicatechin, (−)-epigallocatechin, and (−)-epicatechin-3-O-gallate) with C4–C6 or C4–C8 linkages and monomeric units [51]. Quercetin in the wines was determined in the range of 0.22–0.32 g/L, with sample *A* having a statistically lower content. Similar results were obtained by Budic-Leto et al. [52] in a study on Croatian red dessert wines, where the determined quercetin levels varied from 0.00 to 12.402. In contrast, Panceri et al. [44] obtained slightly higher values ranging from 2.33 to 15.58 g/L. The concentration of flavonols in wines largely depends on grape variety, but also on environmental factors and winemaking practices [52].

The examined wines were characterized by a rather high content of vanillic and syringic acids: 1.38–2.83 g/L and 1.27–2.25 g/L, respectively (Table 5). Avizcuri-Inac et al. [47] reported much lower values—nd.–0.15 mg/L and nd.–0.25 mg/l, and in one case, only 1.41 mg/L. The scatter in the values obtained for these acids in red wines from dried grapes was high. According to Rozdrigez-Delgrado et al. [49] these contents were higher—1.71–2.99 g/L and 1.64–2.77 g/L, and similar to the results obtained in this study. On the other hand, the content of these acids in the experiment of Panceri et al. [44] was significantly higher (i.e., 3.53–13.42 g/L and 1.40–8.85 g/L).

The concentrations of hydroxycinnamic acids determined in wine samples *A*, *B*, and *C* were within typical ranges characteristic of white wines [6,48,52] (Table 5). Wine *B* was distinguished by having the highest content, which, taking into account the maceration process, is the expected result.

A very low content of *trans*-resveratrol (0.03–0.18 mg/L) was determined in the examined wines *A* and *C*, and this stilbene was completely absent in sample *B* (Table 5). Vitalini et al. [53] tested, among others, three Italian dessert wines from dried grapes, and only in Santelmo—Vin Santo DOC did they determine a *trans*-resveratrol content of 0.02 mg/L. A much higher concentration was recorded by Loizzo et al. [35] in Passito of Saracena wine—4.6 mg/L. Classic red wines, according to a large study by Stervbo et al. [54] contained between 0 and 14.3 g/L *trans*-resveratrol. Grapes, and products made from them, are considered a very good source of *trans*-resveratrol, which was attributed as being a potent anti-cancerous compound, although recent studies have not confirmed its unique properties. Numerous analyses of the chemical composition of wines have revealed that these products differ greatly in their *trans*-resveratrol content. It is most often found in red wines, less often in rosé wines, and least often in white wines [53]. The method of beverage production is also of great importance, as also demonstrated in this study. Although the skins of ‘Hibernal’ grapes, despite their golden-pink colour, contain a small amount of *trans*-resveratrol, possibly due to their Seibel 7053 genetic lineage, we would expect the highest content of this compound in sample *B* but due to the use of maceration, it probably degraded during this process for unknown reasons.

A sensory evaluation of the straw wines was carried out, where the respondents assessed their appearance, aroma, flavor, and quality (Table 6). Wines *A* and *B* were rated as wines with an intense amber colour (Figure 2), full flavor, and a long finish. Both in terms of aroma and taste, flavors defined as typical for *passito* wines prevailed (i.e., mainly dried fruit, honey, caramel and nuts [3,5,6]). The high alcohol content of these wines (18%) was fully palpable and also visible through “tears” on the walls of the glass [55]. Of the two wines, wine *B* was rated higher. Wine *C*, in terms of aromas, corresponded to the varietal characteristics, which was reminiscent of Riesling, with a slight addition of vanilla, honey, and linden flowers. The flavor was mainly green fruits, grapefruit, and lemon. This wine was well received and assessed to be very good in terms of quality.

## 4. Conclusions

This study analyzed three wine types from semi-dried grapes using three different methods—classical straw wine *A*, straw wine with fermentative maceration *B*, and refermentation (or Tokaj) *C* methods. They were compared in terms of chemical characteristics, including acid and phenolic profiles, and potential antioxidant properties.

The conducted analyses determined the chemical parameters of wines and their quantitative ranges, after which it was possible to ascertain which of the production methods accumulated the most bioactive compounds. The results of this study showed that the wine produced using the method with seven days maceration had stronger antioxidant and anti-radical properties compared with others.

Sensory analysis showed that wine *B*, despite its high acidity, was the most balanced. The research group selected wine *B* (modification of the passito style with a seven-day maceration of grapes) and wine *C* (Tokaj wine style five Puttonyos) as wines of very good quality. Both the chemical analyses of the wines and their sensory evaluation have demonstrated that the production of *passito* wines from hybrid grape varieties is an effective alternative to the traditional production process and can be successfully applied in cold climates. Future studies will focus on the chemical and sensory changes occurring in wine produced using various techniques during the aging process, combined with a detailed analysis of polyphenols and volatile compounds.

## Figures and Tables

**Figure 1 foods-11-01027-f001:**
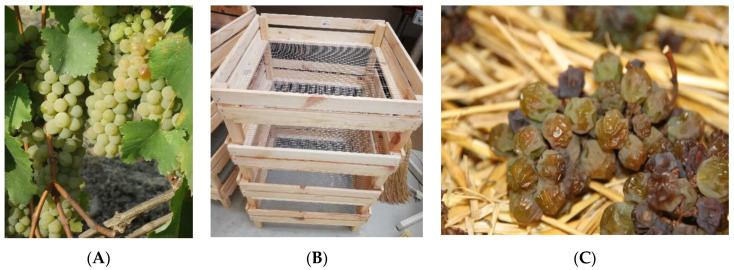
(**A**)—‘Hibernal’ variety; (**B**)—openwork boxes; (**C**)—dehydrated grapes after six weeks.

**Figure 2 foods-11-01027-f002:**
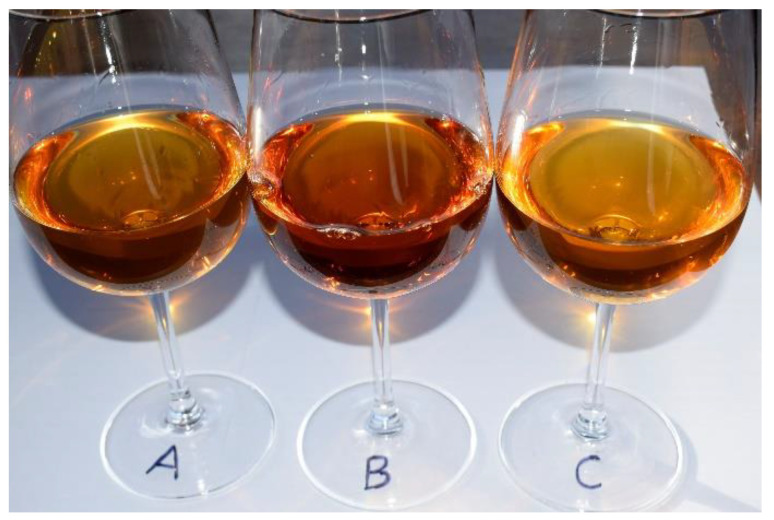
Differences in the colour intensity of wine depending on the production method.

**Table 1 foods-11-01027-t001:** The basic chemical parameters of the grapes used in the experiment: pH, titrable acidity (TA, in tartaric acid equivalents), and the total extract content (TSS).

	Fresh Grapes	Withered Grapes
pH	3.14 ± 0.01	3.31 ± 0.01
Ta (TAE, g/L,)	8.7 ± 0.15	17.9 ± 0.71
TSS (°Brix)	17.63 ± 0.01	40.10 ± 0.00

**Table 2 foods-11-01027-t002:** Organolepic evaluation criteria.

**APPEARANCE**	
Clarity	clear-hazy (faulty?); intensity: pale, medium, deep
Colour	lemon, green, lemon, gold, amber, brown
Other observations	e.g., legs/tears, deposit, petillance, bubbles
**NOSE**	
Condition	clean, unclean (faulty?); intensity: light, medium (−), medium, medium (+), pronounced
Aroma characteristics	e.g., fruits, flowers, spices, vegetables, oak aromas, other
Development	youthful, developing, fully developed, tired/past its best
**PALATE**	
Sweetness	dry, off-dry, medium-dry, medium-sweet, sweet, luscious
Acidity	low, medium (−), medium, medium (+), high
Tannin	low, medium (−), medium, medium (+), high
Alcohol	low, medium (−), medium, medium (+), high
Body	light, medium (−), medium, medium (+), full
Flavour	intensity light, medium (−), medium, medium (+), pronounced
Flavour characteristics	e.g., fruits, flowers, spices, vegetables, oak flavours, other
Length	short, medium (−), medium, medium (+), long
**CONCLUSIONS**	
Quality level	faulty, poor, acceptable, good, very good, outstanding
Level of readiness for drinking/potential for an ageing	can drink now, drink now, not for drinking, too young but has potential, suitable for ageing, too old for ageing, for ageing or further ageing.

**Table 3 foods-11-01027-t003:** Basic chemical characteristics of wines using semi-dried grapes, and which were obtained using three methods of production.

	Wine *A*	Wine *B*	Wine *C*
pH	3.91 a ± 0.18	3.53 b ± 0.01	3.1 c ± 0.01
TA (g/L)	8.18 c ± 0.11	10.34 a ± 0.41	8.95 b ± 0.05
TSS (°Brix)	18.2 a ± 0.00	15.9 b ± 0.06	6.9 c ± 0.00
ethanol (vol %)	18.0 a ± 0.00	18.0 a ± 0.00	15.0 b ± 0.00
tartaric acid (g/L)	2.23 b ± 0.03	4.61 a ± 0.33	2.27 b ± 0.45
citric acid (g/L)	0.50 a ± 0.04	0.55 a ± 0.05	0.28 b ± 0.04
malic acid (g/L)	2.47 b ± 0.10	2.99 a ± 0.19	1.25 c ± 0.10
succinic acid (g/L)	1.60 b ± 0.03	2.40 a ± 0.07	0.57 c ± 0.20
lactic acid (g/L)	0.52 a ± 0.27	0.54 a ± 0.14	0.58 a ± 0.25
acetic acid (g/L)	1.50 a ± 0.01	0.95 b ± 0.01	0.19 c ± 0.01

The values are given as means ± standard deviations, followed by the letters a–c to indicate statistical significance. The values marked with the same letters in one line are not statistically different at α < 0.05; TA—titrable acidity, TSS—total extract content.

**Table 4 foods-11-01027-t004:** The total phenolic content (TPC, mg/L GAE) in straw wines which were obtained by three methods, their antioxidant capacities (FRAP and CuPRAC, both in mmol/L TE), and the radical scavenging capacity (RSC, mmol/L TE).

	Wine *A*	Wine *B*	Wine *C*
TPC	1 082.2 b ± 90.76	3 748.5 a ± 354.74	907.5 b ± 33.23
FRAP	2.4 b ± 0.28	10.2 a ± 1.20	3.4 b ± 0.39
CuPRAC	4.9 b ± 0.22	24.6 a ± 1.66	5.6 b ± 0.42
RSC	1.8 c ± 0.05	5.0 a ± 0.74	1.9 b ± 0.04

The values are given as means ± standard deviations, followed by the letters a–c to indicate statistical significance. The values marked with the same letters in one line are not statistically different at α < 0.05.

**Table 5 foods-11-01027-t005:** Content of selected phenolic compounds in the straw wine obtained by three methods of production.

Phenolic Compound, mg/L	Wine *A*	Wine *B*	Wine *C*
FLAVONOLS			
(+)-catechin	56.91 a ± 10.12	64.29 a ± 3.65	37.82 b ± 0.30
quercetin	0.22 b ± 0.06	0.32 a ± 0.01	0.32 a ± 0.01
HYDROXYBENZOIC ACIDS			
vanilin acid	1.97 b ± 0.30	2.83 a ± 0.32	1.38 b ± 0.32
syringic acid	1.55 c ± 0.20	2.25 a ±0.19	1.27 b ± 0.20
HYDROXYCINNAMIC ACIDS			
chlorogenic acid	0.2 b ± 0.02	0.57 a ± 0.04	0.51 a ± 0.02
caffeic acid	0.04 c ± 0.01	7.10 a ± 0.71	3.43 b ± 0.03
*p*-coumaric acid	1.28 b ± 0.04	3.00 a ± 0.25	1.58 b ± 0.03
*trans*-cinnamic acid	0.28 b ± 0.06	0.43 a ± 0.01	0.26 b ± 0.01
ferulic acid	0.19 c ± 0.03	2.68 a ± 0.13	0.64 b ± 0.04
STILBENES			
*trans*-resveratrol	0.03 b ± 0.003	nd.	0.18 a ± 0.05

The values are given as means ± standard deviations, followed by the letters a–c to indicate statistical significance. The values marked with the same letters in one line are not statistically different at α < 0.05; nd.—not detected.

**Table 6 foods-11-01027-t006:** Sensory evaluation of wines obtained by three methods of production according to “Wine evaluation sheets” based on the WSET materials.

APPERANCE	Wine *A*	Wine *B*	Wine *C*
Clarity	Clear	Clear	Clear
Intensity	Medium to deep	Deep	Medium
Colour	Amber	Intese amber	Amber
Other observations	Tears	Tears	Tears
**NOSE**			
Condition	Clean	Clean	Clean
Intensity	Pronounced	Pronounced	Medium
Aroma characteristics	Almonds, walnuts, golden apples, honey, wood, dried plum	Almonds, walnuts, golden apples, honey, wood, dried plum	Green apples, pear, hay, vanilla, herbs
Development	Developed	Fully developed	Fully developed
**PALATE**			
Sweetness	Medium sweet to sweet	Medium sweet to sweet	Dry to medium dry
Acidity	Medium	Medium (−)	Medium
Alkohol	High	High	Medium
Body	Full	Full	Medium
Flovour intensity	Pronounced	Pronounced	Medium
Flavour characteristics	Pear, jam, caramel, dried fruit	Apple, jam, caramel, dried fruit	Apple, lemon, grape, pear
Lenght	Long	Long	Medium (+)
**CONCLUSIONS**			
Quality level	Good	Very good	Very good
Level of readiness for drinking/potential for an ageing	Can drink now, but has potential for ageing	Can drink now, but has potential for ageing	Can drink now, not suitable for ageing

## Data Availability

The research at the University of Agriculture in Krakow was subvented by the Polish Ministry of Education and Science.

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
