# Peer review of "The Content of Selected Bioactive Compounds in Wines Produced from Dehydrated Grapes of the Hybrid Variety ‘Hibernal’ as a Factor Determining the Method of Producing Straw Wines"

_foods, 2022, doi:10.3390/foods11071027_

Round 1
Reviewer 1 Report
Dear editor and authors,
The presented study “Evaluation of bioactive compounds content and quality in wines produced from dehydrated ‘Hibernal’ grapes grown in a cool climate” investigates the possibility of producing sweet (straw) wines from cold resistant Hibernal grapes using existing or modified techniques. The produced wines were analysed for various wine parameters and their content of bioactive compounds. Furthermore, they were judged by a sensory panel for their appeal. The study possesses some novelty. While the production of straw wines from Hibernal grapes is not uncommon in cooler climate vineries, the introduction of one modified production technique and the assessment and comparison of general wine parameters and bioactive compounds content between the differently produced wines is informative.
General remark:
The title of the publication only captures an aspect of the study, one that is not even majorly accentuated in the manuscript. While it is up to the authors, they could consider adapting the title or dedicating more attention to bioactive compounds content and quality.
The first two paragraphs of the Results section don’t refer to results and fit better in the Introduction.
In all tables, significance letters indicate significant differences per row and not per column. Furthermore, the expression “are not statistically different at < 0.05” lacks the actual parameter: α? P?
It is not reported if statistical testing was performed for the evaluation of “Quality level” in Table 5 and therefore if the difference in quality between the wines was significant.
Specific remarks:
Line 93: “Research concerning the content of various components after wine fermentation process, allows the development of technologies for the production of dessert wines, but also show the practical side of optimizing the technology.” – This sentence is difficult to understand, please clarify.
Line 107: “made of grapes from the vineyard in Krakow” – Please specify which vineyard unless there is only one in Krakow.
Line 113: “The collected fruits were fully rape” – Please correct spelling.
Line 129: “The must was combined with the nevus” – Please explain or substitute “nevus”.
Line 151: “100 mL of wine with the addition of 10 mL of 2 M Ca(OH)2 was distilled at 20°C” – Please verify if distillation happened at 20°C.
Line 198: “… were mixed with X mL of the wine” – It is never specified how much X is.
Line 241: “The scores presented on the sheetswere intended” – Please correct spelling
Line 271: “The initial juice pH in the grapes after drying increased from 3.14 (Tab. 1) to 3.31 (Tab. 2)” – Both refers to Tab. 1.
Line 272ff: “During fermentation, the pH …” – Please refer to Tab. 2.
Line 292: “it was statistically the highest in sample B, and the lowest in sample A, which was probably due to the high initial acid contents in the grapes” – Please elucidate how the initial acid content in the grapes influences the statistically different acidity in the wines.
Line 324: “but obtained the hower sugar concentration” – Please correct spelling.
Line 384: “The stow wines used in the present study” - Please correct spelling.
Line 486: “… and the scatter in the values was large The scatter in the values obtained for these acids in red wines from dried grapes was high” – This seems a duplication.
Line 516: “it is likely to degrade during the process leading to the production of this wine” – Please elucidate why degradation happens especially with this method.
Author Response
Reviewers
The authors thank very much for valuable comments, which certainly contributed to the transparency of this work.
General remark:
The title of the publication only captures an aspect of the study, one that is not even majorly accentuated in the manuscript. While it is up to the authors, they could consider adapting the title or dedicating more attention to bioactive compounds content and quality.
According to the Reviewer suggestion, the title has been changed.
Current title: The content of selected bioactive compounds in wines produced from dehydrated grapes of the hybrid variety "Hibernal" as a factor determining the method of producing straw wines.
The first two paragraphs of the Results section don’t refer to results and fit better in the Introduction.
According to the Reviewer suggestion, the first two lines have been moved to the introduction.
In all tables, significance letters indicate significant differences per row and not per column. Furthermore, the expression “are not statistically different at < 0.05” lacks the actual parameter: α? P?
In the lines: 285, 379, 454 parameter α was added
It is not reported if statistical testing was performed for the evaluation of “Quality level” in Table 5 and therefore if the difference in quality between the wines was significant.
It is difficult to compile statistics for this type of table, therefore the table contains the answer indicated by at least 60% of panelists in the survey. The methodology has been supplemented with this information. Assessment criteria have also been added.
|
APPEARANCE |
|
|
Clarity |
clear-hazy (faulty?); Intensity: pale, medium, deep; |
|
Colour |
lemon, green, lemon, gold, amber, brown; |
|
Other observations |
e.g. legs/tears, deposit, petillance, bubbles |
|
NOSE |
|
|
Condition |
clean, unclean (faulty?); Intensity: light, medium(-), medium, medium(+), pronounced; |
|
Aroma characteristics |
e.g. fruits, flowers, spices, vegetables, oak aromas, other; |
|
Development |
youthful, developing, fully developed, tired/past its best. |
|
PALATE |
|
|
Sweetness |
dry, off-dry, medium-dry, medium-sweet, sweet, luscious; |
|
Acidity |
low, medium(-), medium, medium(+), high; |
|
Tannin |
low, medium(-), medium, medium(+), high; |
|
Alcohol |
low, medium(-), medium, medium(+), high; |
|
Body |
light, medium(-), medium, medium(+), full; |
|
Flavour |
intensity light, medium(-), medium, medium(+), pronounced; |
|
Flavour characteristics |
e.g. fruits, flowers, spices, vegetables, oak flavours, other; |
|
Length |
short, medium(-), medium, medium(+), long; |
|
CONCLUSIONS |
|
|
Quality level |
faulty, poor, acceptable, good, very good, outstanding; |
|
Level of readiness |
can drink now, drink now, not for drinking, / potential for an ageing: too young but has potential, suitable for ageing, too old for ageing, for ageing or further ageing. |
Line 93: “Research concerning the content of various components after wine fermentation process, allows the development of technologies for the production of dessert wines, but also show the practical side of optimizing the technology.” – This sentence is difficult to understand, please clarify.
The sentences have been deleted.
Line 107: “made of grapes from the vineyard in Krakow” – Please specify which vineyard unless there is only one in Krakow.
According to the Reviewer suggestion, „the vineyard in Krakow” was changed to: the vineyard ‘Garlicki Lamus’ located close to Krakow,
Line 113: “The collected fruits were fully rape” – Please correct spelling.
According to the Reviewer suggestion, the expression was corrected to: The collected fruits were fully ripe
Line 129: “The must was combined with the nevus” – Please explain or substitute “nevus”.
According to the Reviewer suggestion, the expression was changed to: The must was combined with the free-run juice
Line 151: “100 mL of wine with the addition of 10 mL of 2 M Ca(OH)2 was distilled at 20°C” – Please verify if distillation happened at 20°C.
The temperature in the lab is automatically set to the desired temperature, during distillation it was set to 20°C.
Line 198: “… were mixed with X mL of the wine” – It is never specified how much X is.
According to the Reviewer suggestion, we have corrected the amount of wine.
Line 241: “The scores presented on the sheetswere intended” – Please correct spelling
According to the Reviewer suggestion, it was corrected to: “The scores presented on the sheets were intended”
Line 271: “The initial juice pH in the grapes after drying increased from 3.14 (Tab. 1) to 3.31 (Tab. 2)” – Both refers to Tab. 1.
According to the Reviewer suggestion, the sentence was changed: The initial juice pH in the grapes after drying increased from 3.14 to 3.31 (Tab. 1).
Line 272ff: “During fermentation, the pH …” – Please refer to Tab. 2.
According to the Reviewer suggestion, reference to table 2 was placed: During fermentation, the pH value increased to 3.91 for sample A, and the statistically lowest and the most desired pH was achieved for sample C (Tab. 2).
Line 292: “it was statistically the highest in sample B, and the lowest in sample A, which was probably due to the high initial acid contents in the grapes” – Please elucidate how the initial acid content in the grapes influences the statistically different acidity in the wines.
The highest acid content in wine B is a result of the effective extraction of organic acids, including phenolic acids, during the maceration. This process has been widely described in the literature. We also observed the highest concentrations of phenolic acids in B wine using HPLC method. For this reason, we have re-written this sentence in the manuscript as:
"Titratable acidity of wines in the experiment was in the range of 8.18–10.34 g/L; it was statistically the highest in sample B, which was probably due to the high initial acid contents in the macerate which was then fermented (Table. 2)".
Line 324: “but obtained the hower sugar concentration” – Please correct spelling.
According to the Reviewer suggestion, it was corrected to: “but obtained the lower sugar concentration”.
Line 384: “The stow wines used in the present study” - Please correct spelling.
According to the Reviewer suggestion, corrected to: “The straw wines used in the present study”.
Line 486: “… and the scatter in the values was large The scatter in the values obtained for these acids in red wines from dried grapes was high” – This seems a duplication.
It was corrected to: [6] reported much lower values: nd.-0.15mg/L and nd.-0.25 mg/l, and in one case only 1.41mg/L. The scatter in the values obtained for these acids in red wines from dried grapes was high.
Line 516: “it is likely to degrade during the process leading to the production of this wine” – Please elucidate why degradation happens especially with this method.
As we know from literature, and as determined in samples A and C of these studies, 'Hibernal' fruits contain trans-resveratrol, albeit in small amounts. Its absence in sample B suggests that it could be degraded during maceration since this stage is characteristic of the production of wine B. Trans-resveratrol is known to be isomerised to its cis form under the influence of UV and light. It is also unstable under alkaline conditions and at high temperature. Here, the maceration was performed at a temperature of 15oC, under the cover, and the final pH of wine B was similar to others. The degradation might be caused by reactions with the decomposition products of other bioactive compounds. It is even more possible as some polyphenols are more sensitive to changes in physical and chemical factors than resveratrol. Unfortunately, we did not deal with the biochemical analysis of macerate in this work therefore we do not know the mechanisms of these processes.
In order not to make this sentence more precise, we suggest replacing the former sentence with a new sentence, as follow:
The skins of 'Hibernal' grapes, despite their golden-pink colour, contain a small amount of trans-resveratrol, possibly due to their Seibel 7053 genetic lineage and we would expected the highest content of this compound in sample B. However, the compounds probably degraded during this maceration process for unknown for us reasons.
Reviewer 2 Report
Well organized and written experimental design. Results are well documented.
Line 105 - consider to use the word variety instead of strain
Line 384 - Please check - Straw wines instead of stow wines?
Line 518 - Table 5. It is not explained how the data of the organoleptic panel of 30 people was analysed. Please clarify.
Author Response
Reviewers
The authors thank very much for valuable comments, which certainly contributed to the transparency of this work.
Line 105 - consider to use the word variety instead of strain
According to the Reviewer suggestion, we corrected into name of variety.
Line 384 - Please check - Straw wines instead of stow wines?
According to the Reviewer suggestion corrected to: “The straw wines used in the present study”
Line 518 - Table 5. It is not explained how the data of the organoleptic panel of 30 people was analysed. Please clarify.
Specific answers were included into the tables. The answer placed in the table represented at least 60% of the same answers. Evaluation criteria were added to the methodology, along with the way the data were analyzed.
|
APPEARANCE |
|
|
Clarity |
clear-hazy (faulty?); Intensity: pale, medium, deep; |
|
Colour |
lemon, green, lemon, gold, amber, brown; |
|
Other observations |
e.g. legs/tears, deposit, petillance, bubbles |
|
NOSE |
|
|
Condition |
clean, unclean (faulty?); Intensity: light, medium(-), medium, medium(+), pronounced; |
|
Aroma characteristics |
e.g. fruits, flowers, spices, vegetables, oak aromas, other; |
|
Development |
youthful, developing, fully developed, tired/past its best. |
|
PALATE |
|
|
Sweetness |
dry, off-dry, medium-dry, medium-sweet, sweet, luscious; |
|
Acidity |
low, medium(-), medium, medium(+), high; |
|
Tannin |
low, medium(-), medium, medium(+), high; |
|
Alcohol |
low, medium(-), medium, medium(+), high; |
|
Body |
light, medium(-), medium, medium(+), full; |
|
Flavour |
intensity light, medium(-), medium, medium(+), pronounced; |
|
Flavour characteristics |
e.g. fruits, flowers, spices, vegetables, oak flavours, other; |
|
Length |
short, medium(-), medium, medium(+), long; |
|
CONCLUSIONS |
|
|
Quality level |
faulty, poor, acceptable, good, very good, outstanding; |
|
Level of readiness/ potential for an ageing |
can drink now, drink now, not for drinking/ too young but has potential, suitable for ageing, too old for ageing, for ageing or further ageing. |
Reviewer 3 Report
The manuscript shows a survey about the impact of three different production methods with regards to the special composition of certain sweet wines.
Before accepting the manuscript changes, clarifications and informative additions have to be made, see below:
- The grape variety Hibernal belongs to a new group of vines that are the results as hybrids of diff. varieties and clones. Aim of the hybrid process is the improvement of fungal resistance in comparison to "classical" grape vines.
- The use of Hibernal or other hybrids is extremely low worldwide. Nevertheless it justifies the research presented in this paper.
- Missing information:
- amount of grapes for each variant, volumes of fermentation musts?
- was SO2 used to stabilize the wines? What was the final concentration in each experimental wine? Was that within the legal limits?
- If SO2 hasn't been used the results are extremely doubtful.
- Correct name of the used yeast strain must be mentioned (dried yeast and yeast producer). What was the rehydration process and the amount of added yeasts?
- nothing is mentioned about the sensory impact of acetic acid. 1.5 g/L must have been noticed by the panelists.
- Last but not least: it would be interesting to see the data of the different musts BEFORE fermentation to see the development and changes due to the fermentation process.
- Typing errors: 55/115/324/Table 5/ etc.
- Concentration of alcohol should be expressed as %vol not in %.
- It should be honestly mentioned that Hibernal is still an "exotic" vine variety that surely will have a future as fungi-resistant grape but is still far away from being noticed by wine consumers.
Author Response
Reviewers
The authors thank very much for valuable comments, which certainly contributed to the transparency of this work.
The grape variety Hibernal belongs to a new group of vines that are the results as hybrids of diff. varieties and clones. Aim of the hybrid process is the improvement of fungal resistance in comparison to "classical" grape vines.
In cold climates, the 'Hibernal' variety is one of the most promising hybrid varieties, due to its aroma and late maturation date, it is suitable for the production of traditional wines and wines produced with special methods.
The use of Hibernal or other hybrids is extremely low worldwide. Nevertheless it justifies the research presented in this paper.
Vitis Vinifera is of the greatest economic importance in the production of wine. In recent years, viticulture has expanded to cooler regions of the world (Central and Eastern Europe). For these geographical regions, hybrid grape varieties are crucial and allow for the production of unique wines from these geographical regions.
Missing information:
- amount of grapes for each variant, volumes of fermentation musts?
For the first two trials, 50 kg of dried grapes were used. In the first test, about 20l of must was squeezed out of 50 kg and fermented. In the second trial, 50 kg of dried grapes were ground and macerated for 7 days. Then the must was poured, the grapes were squeezed out, the liquids were combined (about 20 l) and fermented. In the third test, 3 kg of dried grapes were poured over 20 liters of fresh wine and fermented together. After fermentation, the fresh wine was poured over the grapes and allowed to mature.
- was SO2 used to stabilize the wines? What was the final concentration in each experimental wine? Was that within the legal limits?
Yes, SO2 was obviously used, in concentration of 10g/hL. The final SO2 content of the wine was not checked because the wine was not for sale. However, based on standard wine production, where a similar dosage is used for sweet wines, the allowable SO2 dosage of 400mg/L in the final product was never exceeded.
- If SO2 hasn't been used the results are extremely doubtful
Answer as above.
- Correct name of the used yeast strain must be mentioned (dried yeast and yeast producer). What was the rehydration process and the amount of added yeasts?
The following information was added into methodology:
“Fermentation of the young wine was carried out with ZYMAFLORE® VL1 yeast made by Laffort. IOC Bayanus yeast was used for refermentation at concentration of 50g/hl.”.
“Active dried yeast IOC Bayanus produced by Institut Oenologique de Champagne (Saccharomyces cerevisiae bayanus) was used to produce and referment the wines used in the research. After the supernatant was decanted, the wine was allowed to clarify.”.
- nothing is mentioned about the sensory impact of acetic acid. 1.5 g/L must have been noticed by the panelists.
The acetic acid sensation appeared in the few responses of the panelists and therefore was not included in the description.
- Last but not least: it would be interesting to see the data of the different musts BEFORE fermentation to see the development and changes due to the fermentation process.
The authors agree with the reviewer's comment, unfortunately in the described experiment polyphenolic compounds were not performed before fermentation, but this will be considered in future experiments.
- Typing errors: 55/115/324/Table 5/ etc.
Marked clerical errors have been corrected.
- Concentration of alcohol should be expressed as %vol not in %.
According to the Reviewer suggestion, it was corrected to %vol.
- It should be honestly mentioned that Hibernal is still an "exotic" vine variety that surely will have a future as fungi-resistant grape but is still far away from being noticed by wine consumers.
In the region of Central and eastern Europe, variety Hibernal is of great interest to consumers for its aroma, so it is economically important for this region. However, in the scale of cultivated varieties in the world, Vitis vinifera is a small percentage.